# Influencing Appropriate Statin Use in a Charity Care Primary Clinic

**DOI:** 10.3390/healthcare10122437

**Published:** 2022-12-02

**Authors:** Hasitha Diana Manohar, Carole Karkour, Rajesh N. Desai

**Affiliations:** 1Department of Internal Medicine, Loyola University Medical Center, Maywood, IL 60153, USA; 2Department of Internal Medicine, Wood Johnson Medical School, Saint Peter’s University Hospital and Rutgers Robert, New Brunswick, NJ 08901, USA

**Keywords:** appropriate statin use, electronic medical records (E.M.R.), management of cholesterol, level of providers, pharmacy levels

## Abstract

According to the American College of Cardiology/American Heart Association (ACC/AHA) new cholesterol management guidelines in 2019, statin regimen was prescribed to only about 46.4% and 30% of diabetes (DM) patients and patients with atherosclerotic cardiovascular disease (ASCVD), respectively. Atherosclerotic cardiovascular disease accounts for most deaths and disabilities in North America. This study argues that a systematic approach to identifying targeted interventions to adhere to the statin regimen for ASCVD is sparse in previous studies. This study seeks to address the research gap. Besides, the study argues that the statin regimen could improve cholesterol management with the enablers of pharmacy, providers, electronic medical records (E.M.R.), and patients. It paves the way for future research on cardiovascular and statin regimens from different perspectives. Current study has adopted the Qualitative observation method. Accordingly, the study approached the charity care primary clinic serving a large population in the northeastern part of the United States, which constitutes the project’s setting. The facility has 51 internal medicine residents. The facility has E.H.R., which is used by the clinical staff. Besides, providers use electronic medication prescribing (E-Scribe). Four PDSA cycles were run in six months. Here, the interventions were intensified during each subsequent cycle. The interventions were then incorporated into routine clinical practice. Based on the observation, the study found a 25% relative improvement by six months based on the baseline data of the appropriate intensity statin prescription for patients with ASCVD or DM by medical resident trainees in our single-center primary care clinic. A total of 77% of cardiovascular disease patients were found to be on an appropriate statin dose at baseline. On the other hand, the proportion of patients with DM who were on proper dose statin was 80.4%. According to the study’s findings, PDSA could result in a faster uptake and support of the ACC/AHA guidelines. Evidence indicates that overmedication of persons at low risk and time constraints are some of the most significant impediments to the greater use of prescription medications. Proactive panel management can help improve statins’ use by ensuring they are used appropriately.

## 1. Public Interest Summary

Evidence-based measurements are becoming increasingly popular among policymakers and administrators in the healthcare sector to identify shortcomings in healthcare delivery. As a result, it is critical to characterize health inequalities as discrepancies in the quality of care provided. This has the potential to provide an effective framework for those who are involved in initiatives to identify health disparities. Furthermore, it is most successful in leveraging performance improvement when evidence-based measures are used. Therefore, studies focused on inequalities are extremely important for policymakers to consider. In order to address this issue, some researchers are advocating for introducing quality measures suitable for tracking healthcare quality discrepancies at the subpopulation level.

## 2. Background

Approximately 945,000 deaths are attributed to cardiovascular diseases annually in the United States, which accounts for about 41% of all mortalities in the country [1]. Cardiovascular diseases also contribute to high morbidity in the country. Indeed, they account for about 6 million hospital admissions annually [2,3]. Diabetes mellitus refers to dysglycaemia that is accompanied by nephropathy, retinopathy, and cardiovascular diseases, among others. Some cardiovascular diseases associated with diabetes mellitus include congestive heart failure, stroke, cardiomyopathy, and coronary heart disease, among others. Cardiovascular disease and diabetes mellitus comorbidities are associated with early death. In 1999, a collective effort of the following organizations resulted in recognition of the need for collective cooperation in the prevention of cardiovascular disease in diabetes patients: the American Diabetes Association (A.D.A.), the National Institute of Diabetes and Digestive and Kidney Diseases, and National Heart, Lung, and Blood Institute (NHLBI), Juvenile Diabetes Foundation International, and the American Heart Association (A.H.A.) [4]. The pathophysiology of cardiovascular disease involves the thickening of arterial walls, which occurs gradually.

Clinical detection of atherosclerosis is only possible at an advanced level, which has made clinical assessment impractical. However, imaging technology advancements have produced sophisticated and effective tools for detecting atherosclerosis at an early stage. These advancements have also made it possible to classify people based on their exposure to the risk of atherosclerosis symptoms. These advancements also provide the means for assessing the treatment results and improving the current understanding of atherosclerosis biology [5]. The problem statement of the present study is as follows. The American College of Cardiology/American Heart Association (ACC/AHA) released new cholesterol management guidelines in 2019. Since then, the guidelines have been updated severally. Indeed, the guidelines are updated frequently to incorporate new evidence. However, in many cases, healthcare providers do not keep up with the frequency at which changes are made.

Consequently, they often fail to implement them in clinical practice. Based on the latest A.H.A. data, an appropriate statin regimen was prescribed to only about 46.4% and 30% of diabetes patients and patients with atherosclerotic cardiovascular disease (ASCVD). Charity care clinics can adopt various interventions for safe statin use. Such interventions include the following: educational programs, frequent reminders, support tools, and outcome evaluation [6].

Healthcare providers also need to educate their patients on the right dose and side effects of statins. They should also regularly check the patient’s medication list to ensure safety and prevent drug–drug interaction. Nursing staff also need to verify medication compliance and conduct patient counseling on various issues, such as the recommended way to administer drugs and report any possible identified concerns to prescribers. Efforts should also be made to ensure equal access to the same information among members of interprofessional teams for informed decision-making and improved therapeutic outcomes. Consequently, we present a hypothesis that an approach that involves systematic identification of targeted interventions and barriers at the following levels presents high prospects of improving cholesterol management efforts: electronic medical records (E.M.R.), pharmacy levels, and providers. 

Atherosclerotic cardiovascular disease accounts for most deaths and disabilities in North America. In response to this high burden, the American Heart Association launched an initiative in 1996 dubbed “Guide to the Primary Prevention of Cardiovascular Disease.” The guide was updated in 2002 [7]. These guidelines, however, do not address the prevention of atherosclerosis in children. However, interventions targeting this group carry many prospects of success [8]. 

## 3. Methodology

To achieve a 25% relative improvement by six months based on the baseline data of the appropriate intensity statin prescription for patients with ASCVD or D.M. by medical resident trainees in our single-center primary care clinic.

### 3.1. Family of Measures

In Table 1, it’s the conceptual definition of our study measures.

### 3.2. Key Stakeholders

People involved in statin use in the charity care primary clinic are stakeholders. The following matrix explains the role of interest and power in the high and low context (Figure 1). When interest and power are high, their players (physician providers/ Medical resident trainees, ACC/AHA guidelines) play a vitally important role in the matrix. When the role of interest and power is low, the crowd (pharmacy, panel management, clerical staff) in the matrix plays a vibrant role. When the role of interest is high and power is low, the context setters (Administrative leadership, E.M.R., Educational sessions, and online questionnaires) in the matrix play a dynamic role. When the role of interest is low and power is high, the subjects (Patients, caregivers, delivery system reform incentive payment program (DSRIP)) play a vicarious role. 

At the beginning stage of the research, a questionnaire was circulated among 51 internal medicine residents to identify the barriers to appropriate statin prescriptions at the clinic. Ten items of the questionnaire were measured using the dichotomous scale (Yes/No) given in Table 2. Items of the questionnaire were adopted from the poster by Khetan et al. in March 2017. The details of the participants were kept anonymous in order to avoid social desirability bias among the participants of the study. 

4c PDSA cycles were run during the study’s observation for six months. The PDSA cycle is an iterative, four-step model for improving a process. The first step is developing a plan in which predictions of outcomes are clearly stated and tasks are assigned. The plan’s who, what, when, and where are decided in this phase. In the “do” phase, the plan is implemented. Data and results obtained are then analyzed in the “study” phase. Lastly, the plan is either adopted, adapted, or abandoned in the “act” phase based on the evaluation of the data in the prior step. Learning from one cycle should guide the cycles that follow. Here, the Interventions intensified during each subsequent cycle. The interventions were then incorporated into routine clinical practice. 

The present study intended to improve the performance of our institutional practice. The study also constitutes a part of resident education. Consequently, it did not go through the I.R.B. review process. Additionally, the patient data were obtained from the data bank available for hospital quality metrics.

### 3.3. Data Analysis

#### 3.3.1. Problem Characterization 

Characterization of the problem for an appropriate statin prescription could be broadly classified into four categories (see Figure 2). They are (a) provider, (b) patient, (c) EMR, (d) Pharmacy. “Provider” barriers could be broadly classified into six major points: a priority, lack of awareness, fear of adverse effects, disagreement, fear of overtreatment, and time constraints. “Patient” barriers could be classified into four major points: the worry of side effects, costs, lack of interest in prevention, and under-appreciation of risk. THE “E.M.R.” barrier relies on no risk calculator/template. The “Pharmacy” barrier relies on no pharmacy chart. 

#### 3.3.2. Intervention

Medical resident trainees in the clinic were surveyed to understand the barriers to prescribing statins and are identified. 

#### 3.3.3. Implementation

In Table 3, it shows the PDSA cycles that we implemented and the outcomes obtained.

## 4. Findings

Seventy-point seven percent of cardiovascular disease patients were found to be on the appropriate statin dose at baseline. On the other hand, the proportion of patients with diabetes mellitus who were on proper dose statin was 80.4%. After one month, the proportion of patients with cardiovascular disease on the appropriate statin dose had increased by 8.4%. On the other hand, the proportion of patients with diabetes mellitus who were on appropriate statins increased by 4.2%. Figure 3 shows the results after six months. 

## 5. Limitations

A fair change in clinical practice was observed among the internal medicine resident community, though the aim was not achieved by 25% relative improvement. In addition, an attempt to compare the prescription of statin use among the attendings and residents was made but did not receive adequate responses to the survey among the attending population.

## 6. Discussion

In March 2019, a new guideline for the initiation of statin treatment targeting cardiovascular disease prevention was released by the American Heart Association (ACC/AHA) and the American College of Cardiology [9]. Unlike the previous guidelines, these guidelines focused on the risk of total atherosclerotic cardiovascular disease (ASCVD) [10,11,12]. In this case, the cardiovascular disease risk employed in these guidelines was based on deemphasizing low-density lipoprotein (LDL) cholesterol thresholds and new Pooled Cohort Equations. Four categories of statin treatment eligibility for adults aged 40–75 years were established by the ACC/AHA guidelines. The new cardiovascular disease risk threshold provides the basis for the use of statins in treating abnormal cholesterol in 8.2 million adults more in the United States [13]. However, expanding the eligible age for starting statin treatment under the ACC/AHA guidelines has elicited controversy [14]. Some critics have argued that the Pooled Cohort Equations are prone to overestimating risk. This exposes many adults to unnecessary statin use in the United States [15,16]. Optimal Medline **search strategies** can be developed and tested by **retrieving** sound clinical **studies** on prevention or **treatment** of health disorders [17,18,19].

Statins are well tolerated. However, recent evidence suggests that it could lead to an increased risk of diabetes [7,20]. However, there is no consensus on this observation. Instead, other researchers suggest that statins should be used under ACC/AHA guidelines based on the conclusion that statins reduce the risk of LDL cholesterol [7,16,21]. In previous studies, evidence shows that using a relatively lenient threshold could be cost-effective. This is the case even though statin use is associated with diabetes risk [22]. Our studies seek to generate a hypothesis to aid in the analysis of cost-effectiveness based on the ACC/AHA guidelines. This effort stands to help in establishing optimal value for cardiovascular disease threshold for ten years [13,23,24]. These healthcare use disparities are potentially attributed to factors linked to providers, patients, or systems [25]. Alternatively, they could be attributed to social inequalities such as socioeconomic status. Consequently, there is a need for healthcare researchers to understand and examine possible ways in which the underlying factors (at the system, organizational, individual, and provider levels) impact health and disparities in healthcare status and access. 

Policymakers and administrators in the healthcare sector are increasingly adopting the use of evidence-based measures to detect inadequacies in healthcare quality. The importance of clinical decision-making can be seen in the clinical reasoning which is based on complex and multifaceted cognitive processes, and the level of cognition is perhaps the most relevant factor that impacts the physician’s clinical reasoning [26,27,28]. Therefore, it is vital to define health disparities as gaps in the quality of care. This offers a practical framework for efforts to detect health disparities. Furthermore, evidence-based measures are mainly effective in leveraging performance improvement [29]. Consequently, studies focusing on disparities are highly necessary for policymakers. Therefore, some researchers push for developing quality measures to monitor healthcare quality disparities at the subpopulation level [30]. 

This study further highlights the significance of the role played by the leadership of junior doctors in the Implementation and sustenance of the improvement programs. Junior doctors remain untapped as a potential talent pool for quality improvement measures. Therefore, there is a need to equip them with information on potential pitfalls, which can be instrumental in reducing system errors that commonly inflict healthcare organizations [31]. There is a need for the e-learning platform to improve the performance outcome of these junior doctors [32]. Through teamwork and effective communication, the consistency of the PDSA cycle was realized. Efforts to demonstrate concurrent maintenance of morale change implementation poses a challenge. However, it is necessary for success [33].

## 7. Conclusions

In conclusion, this study finds that PDSA cycles pose significant prospects for increased rates of statin use. In addition, the study finds that PDSA could lead to quicker ACC/AHA guideline uptake.

## 8. Implications

Insights that have already been gathered show that doubts about the guidelines, overmedication of people at low risk, time limitations, and time limitations constitute some of the main barriers to increased use of prescriptions. Enhancing appropriate usage of statin can be improved through proactive panel management. 

## 9. Key Highlights

The experiment takes place in a charity care clinic serving a large population in the northeastern United States, serving a huge population. There are 51 internal medicine residents at the facility. The facility has an electronic health record system used by the clinical staff.

In addition, clinicians use electronic drug prescribing to streamline the process (E-Scribe). 

This quality improvement project hypothesizes that an education campaign on electronic pop-up reminders and evidence-based algorithms targeting prescribers will have the capacity to increase awareness of cardiovascular disease and diabetes mellitus recommendations in the future.

*This study result was presented at the following conferences*:2021 Annual Research Day, Saint Peter’s University Hospital, New Brunswick, United States.ACPNJ Scientific meeting 2020, Princeton, United States.Resident 360 QI Challenge 2020.

## Figures and Tables

**Figure 1 healthcare-10-02437-f001:**
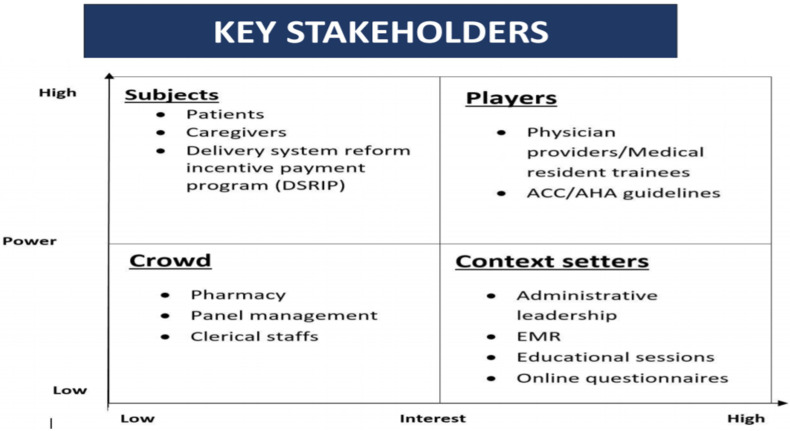
Illustration of key stakeholders in the order of interest in the study and power to contribute to the study.

**Figure 2 healthcare-10-02437-f002:**
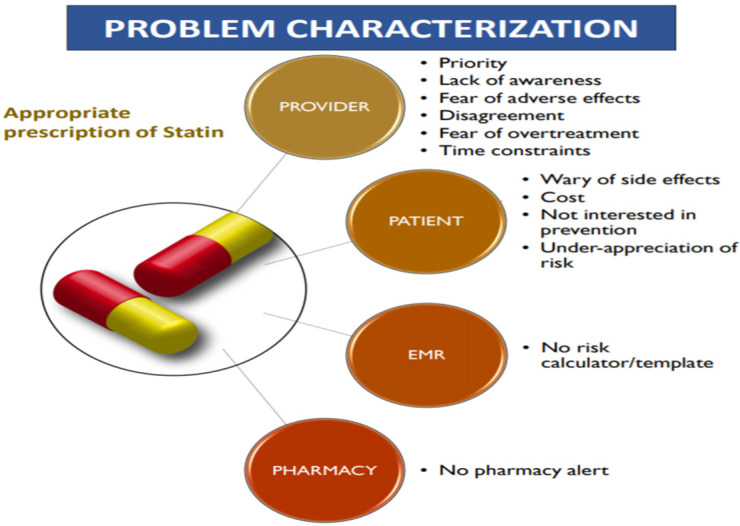
Pictorial representation of the various factors affecting the appropriate prescription of statins.

**Figure 3 healthcare-10-02437-f003:**
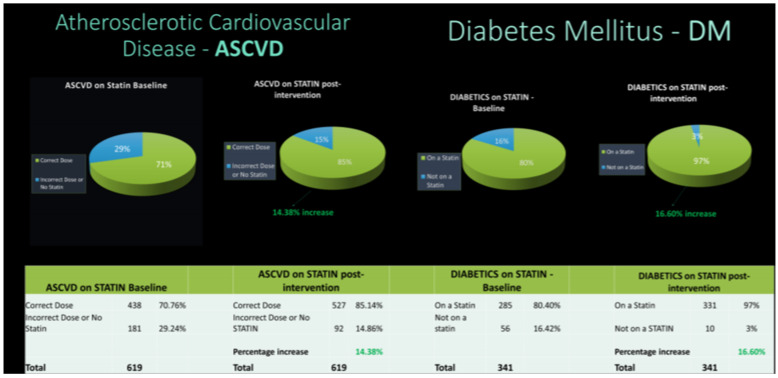
Comparison of appropriate statin use in both groups at baseline and the end of the study.

**Table 1 healthcare-10-02437-t001:** Conceptual definition of our study measures.

Measure	Conceptual Definition
**Outcome**	Information regarding statin use among patients who visited the clinic, aged between 40–75 years.
**Process**	PowerPoint presentations on current statin guidelines, educational sessions, panel management.
**Balancing**	Statins discontinued due to significant side effects.
**Qualitative**	Appropriate statin usage at the end of 6 months. Resident knowledge of proper statin use.

**Table 2 healthcare-10-02437-t002:** Resident survey questionnaire and results.

Survey Questions	Count of “Yes”	Total	% of “Yes”
Worry about overmedicating people at low risk	30	49	61.22
Patient Preference	21	49	42.86
Time illumination	14	49	28.57
The risk of Prescribing outweighs the benefit	11	49	22.45
Unsure as to whom to use the calculator	10	49	20.41
I do not know the guidelines	10	49	20.41
Do not prescribe a statin to simplify the regimen for the patient	9	49	18.37
Guidelines are too complicated	8	49	16.33
Unsure about the clinical benefit	8	49	16.33
Disagree with the guidelines	6	49	12.24

**Table 3 healthcare-10-02437-t003:** PDSA cycles that we implemented and the outcomes obtained.

PDSA	Cycle Description	Lessons Learned
**1**	PowerPoint presentations on current statin guidelines	Agreement with guidelines
**2**	Educational sessions	Not worrying about over-medicating people at low risk
**3**	Panel Management	Proactive panel management to ensure up-to-date primary preventive care.
**4**	E.M.R. generated a list of patients who were not on appropriate statin	Gaps in knowledge and practice.

## Data Availability

Not applicable.

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
