# Peer review of "Influencing Appropriate Statin Use in a Charity Care Primary Clinic"

_healthcare, 2022, doi:10.3390/healthcare10122437_

Round 1

Reviewer 1 Report

Abstract:

·         The whole abstract needs to be reviewed. Sentences are not under the appropriate headings, for example under objective the authors explain the study setting (this should be under methods). It’s not clear what is the objectives of the study. The abstract need to be clearly written. In addition, some abbreviations such as PDSA, ACC, AHA are not defined (same in the manuscript body).

Methods:

·         Some detail on the questionnaire is required; for example; how many items, responses type, how it was developed? Etc.

·         How many cycles of PDSA were conducted? During how many months?

·         Was it possible to apply for ethical approval?

·         What was the objectives of knowing the barriers to appropriate statins prescribing? Was the result useful for the implementation step?

Results:

·         Have you identified any factors (demographics or other) that might be associated inappropriate statins prescribing?

·         Was the result stratified by types of statins?

·         I understand that this is a quality improvement project but any further detail about the results could be helpful.

Discussion:

·         The discussion should focus more on justification of the study limitation and comparison with similar studies and quality improvement projects.

References

References are up to date.

Author Response

Reviewer 1- Comments

Comment 1: The whole abstract needs to be reviewed. Sentences are not under the appropriate headings, for example under objective the authors explain the study setting (this should be under methods). It’s not clear what is the objectives of the study. The abstract need to be clearly written. In addition, some abbreviations such as PDSA, ACC, AHA are not defined (same in the manuscript body).

Author’s Response

Abstract: Although a significant number of researches, findings and advancements are located in the treatment strategies for cardiovascular diseases. Its insights into the risk factors of cardiovascular diseases have been evolving over time. Correspondingly, Cardiovascular diseases are still among the leading causes of death in the developed world. According to the reports of the American College of Cardiology/ American Heart Association (ACC/AHA) released new cholesterol management guidelines in 2019, statin regimen was prescribed to only about 46.4% and 30% of diabetes patients and patients with atherosclerotic cardiovascular disease (ASCVD). While Atherosclerotic cardiovascular disease accounts for most deaths and disabilities in North America. This study argues that a systematic approach to identifying targeted interventions to subscribe to the statin regimen to the ASCVD is sparse in the previous studies. This study seeks to address the research gap. Besides, study argues that the Statin regimen could result in improvement in cholesterol management efforts with the enablers of pharmacy, providers, electronic medical records (EMR), and patients. This paves the way for more future research on cardiovascular and statin regimens from different perspectives.

Objective: The main objective of the study is to test the clinical hypothesis if an education program on electronic pop-up reminders and the evidence-based algorithm has the potential to improve awareness of statin prescription in cardiovascular disease and diabetes mellitus guidelines of targeting prescribers.

Methods: Current study has adopted the Qualitative observation method. Accordingly, the study approached the charity care primary clinic serving a large population situated in the northeastern part of the United States constitutes the setting of the project. The facility has 51 internal medicine residents. The facility has EHR which is used by the clinical staff. Besides, providers use electronic medication prescribing (E-Scribe).

Results: Based on the observation, study found a 25% relative improvement by six months based on the baseline data of the appropriate intensity statin prescription for patients with ASCVD or DM by medical resident trainees in our single-center, primary care clinic.  Seventy-point seven percent of cardiovascular disease patients were found to be on appropriate dose of statin at baseline. On the other hand, the proportion of patients with diabetes mellitus who were found to be on proper dose statin was 80.4%. After one month, the proportion of patients with cardiovascular disease on appropriate dose of statin had increased by 8.4%. On the other hand, the proportion of patients with diabetes mellitus who were on appropriate dose statin increased by 4.2%.

Conclusion: The findings of this study indicate that PDSA cycles have a significant potential for increasing the rates of statin use. According to the findings of the study, PDSA could result in a faster uptake and support of the ACC/AHA guidelines. Evidence gathered indicates that overmedication of persons at low risk and time constraints are some of the most significant impediments to greater use of prescription medications. Proactive panel management can help improve statins’ appropriate use by ensuring that they are used appropriately.

The above responses are precisely provided in the Abstract 

We thank the reviewer for their valuable suggestions and observations.

Comment 2: Some detail on the questionnaire is required; for example; how many items, responses type, how it was developed? Etc.

Author’s Response

At the beginning stage of the research, a questionnaire was circulated among 51 internal medicine residents to identify the barriers to appropriate statin prescription at the clinic. Items of the questionnaire were measured using the dichotomous scale (Yes/No) given in Table 2. Items of the questionnaire were adopted from the poster by Khetan et al in March 2017. The details of the participants were kept anonymous in order to avoid social desirability bias among the participants of the study.

The above responses are precisely provided in 3.2. key stakeholder’s 2nd para (1st half) and Table 2 Survey Questionnaire and Results.  

We thank the reviewer for their valuable suggestions and observations.

Comment 3: How many cycles of PDSA were conducted? During how many months?

Author’s Response

7 PDSA cycles were run during the observation of the study for a period of six months. The PDSA cycle is an iterative, four-step model for improving a process. The first step is the development of a plan in which predictions of outcomes are clearly stated and tasks are assigned. It is in this phase that the who, what, when, and where of the plan is decided. In the “do” phase, the plan is implemented. Data and results obtained are then analyzed in the “study” phase. Lastly, the plan is either adopted, adapted, or abandoned in the “act” phase based on the evaluation of the data in the prior step. The learning from one cycle should guide the cycles that follow. Here, the Interventions intensified during each subsequent cycle. The interventions were then incorporated into routine clinical practice.

The above responses are precisely provided in 3.2. key stakeholder’s 2nd para (2nd half)

We thank the reviewer for their valuable suggestions and observations

Comment 4: Was it possible to apply for ethical approval?

Author’s Response

            No, ethics approval was not sought as this study was performed as a part of the hospital quality monitoring and  improvement project. Any information from the patients charts was collected retrospectively by IT generated lists. Patient identifiers were not used.

We thank the reviewer for their valuable suggestions and observations

Comment 5: What was the objectives of knowing the barriers to appropriate statins prescribing? Was the result useful for the implementation step?

Author’s Response

The main objective of knowing barrier in the research is to potentially address the issue effectively in a practical way. Understanding the barriers involved in the four categories such as (a) provider, (b) patient, (c) EMR, (d) Pharmacy, helps to achieve the appropriate statin prescription at operative implementation level in the clinical setting.

We thank the reviewer for their valuable suggestions and observations

Comment 6: Have you identified any factors (demographics or other) that might be associated inappropriate statins prescribing?

Author’s Response

Study has collectively held the information about the respondents since the main aim of the study is to observe the impact of knowledge on statin use in atherosclerotic cardiovascular disease (ASCVD). Current study has not extended it on the demographic levels. Current findings of the study show that the Seventy seven percent of cardiovascular disease patients were found to be on appropriate dose of statin at baseline. On the other hand, the proportion of patients with diabetes mellitus who were found to be on proper dose statin was 80.4%. After one month, the proportion of patients with cardiovascular disease on appropriate dose of statin had increased by 8.4%. On the other hand, the proportion of patients with diabetes mellitus who were on appropriate does statin increased by 4.2%. 

The above responses are precisely provided in the 4. Findings.

We thank the reviewer for their valuable suggestions and observations

  •  

Comment 7: Was the result stratified by types of statins?

Author’s Response

            Although there are seven types of statins, Atorvastatin, Fluvastatin, Lovastatin, Pravastatin, Rosuvastatin, Simvastatin, Pita vastatin. Current study focused on any statin use to check its impact on the atherosclerotic cardiovascular disease (ASCVD).

We thank the reviewer for their valuable suggestions and observations

Comment 8: I understand that this is a quality improvement project but any further detail about the results could be helpful.

Author’s Response

            Following are the implications of the current study which will help to understand the results of the study at the practical implementation level.

Policymakers and administrators in healthcare sector are increasingly adopting the use of evidence-based measures to detect inadequacies in healthcare quality. Therefore, it is important to define health disparities as gaps in quality of care. This stands to offer effective framework for involved in efforts to detect health disparities. The use of evidence-based measures is mainly effective in leveraging performance improvement. Consequently, studies focusing on disparities are highly necessary for policymakers. Some researchers, therefore, push for the development of quality measures that are suited for monitoring healthcare quality disparities at the subpopulations level.

This study further highlights the significance of the role played by the leadership of junior doctors in the implementation and sustenance of the improvement programs. Junior doctors as potential talent pool for measures of quality improvement remains untapped. Therefore, there is a need for equipping them with information on potential pitfalls, which can be instrumental in reducing system errors that commonly inflict healthcare organizations. Through teamwork and effective communication, the consistency of PDSA cycle was realized. Efforts to demonstrate concurrent maintenance of morale change implementation poses a challenge. However, it is necessary for success.

The above responses are precisely provided in the 3rd and 4th para of the discussion. 

We thank the reviewer for their valuable suggestions and observations

  • Comment 9: The discussion should focus more on the justification of the study limitation and comparison with similar studies and quality improvement projects.

Author’s Response

In the discussion part, current study focused on the explaining the statistical data along with the impacts of the statin regimen on the atherosclerotic cardiovascular disease (ASCVD) and explained the policy implications with focused towards the quality improvement at the practical implementation level. Limitations of the study has given as a separate sub chapter.

In March 2019, a new guideline for the initiation of statin treatment targeting cardiovascular disease prevention was released by the American Heart Association (ACC/AHA) and the American College of Cardiology. Unlike the previous guidelines, these guidelines focused on the risk of total atherosclerotic cardiovascular disease (ASCVD). The cardiovascular disease risk employed in these guidelines, in this case, were based on deemphasizing low-density lipoprotein (LDL) cholesterol thresholds and new Pooled Cohort Equations. Four categories of statin treatment eligibility for adults aged 40-75 years were established by the ACC/AHA guidelines. The new cardiovascular disease risk threshold provides the basis for the use of statin in treating abnormal cholesterol in 8.2 million adults more in the United States [13]. However, the expansion of the eligible age for starting statin treatment under the ACC/AHA guidelines have elicited controversy. Some critics have argued that the Pooled Cohort Equations are prone to overestimation of risk. This exposes many adults to unnecessary statin use in the United States.

Statins are well tolerated. However, evidence gathered recently suggests that it could lead to increased risk of diabetes. However, there is no consensus on this observation. Instead, other researchers suggest that statins should be used under ACC/AHA guidelines based on the conclusion that statins reduce the risk of LDL cholesterol. In previous studies, evidence gathered shows that the use of relatively lenient threshold could be cost-effective. This is the case even though statin use is associated with diabetes risk. Our studies seek to generate hypothesis to aid in the analysis of cost-effectiveness based on the ACC/AHA guidelines. This effort stands to help in establishing optimal value for cardiovascular disease threshold for a period of 10 years. These healthcare use disparities are potentially attributed to factors linked to providers, patients or systems. Alternatively, they could be attributed to social inequalities such as socioeconomic status. Consequently, there is a need for healthcare researchers to understand and examine possible ways in which the underlying factors (at the system, organizational, individual, and provider level) impact health and disparities in healthcare status and access.

Policymakers and administrators in healthcare sector are increasingly adopting the use of evidence-based measures to detect inadequacies in healthcare quality. Therefore, it is important to define health disparities as gaps in quality of care. This stands to offer effective framework for involved in efforts to detect health disparities. The use of evidence-based measures is mainly effective in leveraging performance improvement. Consequently, studies focusing on disparities are highly necessary for policymakers. Some researchers, therefore, push for the development of quality measures that are suited for monitoring healthcare quality disparities at the subpopulations level.

This study further highlights the significance of the role played by the leadership of junior doctors in the implementation and sustenance of the improvement programs. Junior doctors as potential talent pool for measures of quality improvement remains untapped. Therefore, there is a need for equipping them with information on potential pitfalls, which can be instrumental in reducing system errors that commonly inflict healthcare organizations. Through teamwork and effective communication, the consistency of PDSA cycle was realized. Efforts to demonstrate concurrent maintenance of morale change implementation poses a challenge. However, it is necessary for success.

The above responses are precisely provided in the Discussion and Limitations part.

We thank the reviewer for their valuable suggestions and observations

Comment 10: References are up to date.

Author’s Response

            Yes, study has incorporated the recent references

We thank the reviewer for their valuable suggestions and observations

Reviewer 2 Report

The manuscript is very confused, with many flaws in scientific language and organization. The introduction talks about several things and it is not clear what is the focus, there is no rationale for the work to be developed. The objectives are not clear and there is no adequate description of the methods. I was left without understanding what really was designed and developed the work and how it was designed. The results of a questionnaire are presented in the Methods section and also some figures do not make sense and seem to be print screens. 

Author Response

Comment 1: The introduction talks about several things and it is not clear what is the focus, there is no rationale for the work to be developed.

Author’s Response

Approximately 945,000 deaths are attributed to cardiovascular diseases annually in the United States, which accounts for about 41% of all mortalities in the country. Cardiovascular diseases also contribute to high morbidity in the country. Indeed, they account for about 6 million hospital admissions annually. Diabetes mellitus refers to dysglycaemia that is accompanied by nephropathy, retinopathy, and cardiovascular diseases, among others. Some of the cardiovascular diseases that are associated with diabetes mellitus include congestive heart failure, stroke, cardiomyopathy, and coronary heart disease, among others. cardiovascular disease and diabetes mellitus comorbidities is associated with early death. In 1999, a collective effort of the following organizations resulted in the recognition of the need for collective cooperation in the prevention of cardiovascular disease in diabetes patients: the American Diabetes Association (ADA), the National Institute of Diabetes and Digestive and Kidney Diseases, and National Heart, Lung, and Blood Institute (NHLBI), Juvenile Diabetes Foundation International, and the American Heart Association (AHA). The pathophysiology of cardiovascular disease involves the thickening of arterial walls, which takes place gradually.

Clinical detection of atherosclerosis is only possible at advanced level, which has made the use of clinical assessment impractical. However, advancement in imaging technology has produced sophisticated and effective tools for detecting atherosclerosis at an early stage. These advancements have also made it possible to classify people on the basis of their exposure to risk of atherosclerosis symptoms. these advancements also provide the means for assessing the treatment results and improving the current understanding of atherosclerosis biology. The problem statement of the present study is as follows. The American College of Cardiology/American Heart Association (ACC/AHA) released new cholesterol management guidelines in 2019. Since then, the guidelines have been updated severally. Indeed, the guidelines are updated frequently to incorporate new evidence. However, in many cases, healthcare providers do not keep up with the frequency at which changes are made. Consequently, they often fail to implement them in clinical practice.  Based on the latest AHA data, appropriate statin regimen was prescribed to only about 46.4% and 30% of diabetes patients and patients with atherosclerotic cardiovascular disease (ASCVD). Charity care clinics can adopt various interventions for safe statin use. Such interventions include the following: educational programs, frequent reminders, support tools, and outcome evaluation.

There is also a need for healthcare providers to educate their patients on the right dose and side effects of statins. They should also regularly check the list of medications that the patients are taking to ensure safety and prevent drug-drug interaction. There is also a need for nursing staff to verify medication compliance and conducting patient counselling on various issues such as the recommended way to administer drugs and report any possible identified concern prescribers. Efforts should also be made to ensure equal accessibility of same information among members of interprofessional teams for informed decision-making and improved therapeutic outcomes. Consequently, we present a hypothesis that an approach that involves systematic identification of interventions that are targeted and barriers at the following levels present high prospects of improving cholesterol management efforts: electronic medical records (EMR), pharmacy levels, and providers.

The main objective of the study is to test the clinical hypothesis if an education program on electronic pop-up reminders and the evidence-based algorithm has the potential to improve awareness of cardiovascular disease and diabetes mellitus guidelines of targeting prescribers. Current study has adopted the Qualitative observation method. Accordingly, the study approached the charity care primary clinic serving a large population situated in the northeastern part of the United States constitutes the setting of the project. The facility has 51 internal medicine residents. The facility has EHR which is used by the clinical staff. Besides, providers use electronic medication prescribing (E-Scribe).

Although a significant number of researches, findings and advancements are located in the treatment strategies for cardiovascular diseases. Its insights into the risk factors of cardiovascular diseases have been evolving over time. Correspondingly, Cardiovascular diseases are still among the leading causes of death in the developed world. According to the reports of the American College of Cardiology/ American Heart Association (ACC/AHA) released new cholesterol management guidelines in 2019, statin regimen was prescribed to only about 46.4% and 30% of diabetes patients and patients with atherosclerotic cardiovascular disease (ASCVD). While Atherosclerotic cardiovascular disease accounts for most deaths and disabilities in North America. This study argues that a systematic approach to identifying targeted interventions to subscribe to the statin regimen to the ASCVD is sparse in the previous studies. This study seeks to address the research gap. Besides, study argues that the Statin regimen could result in improvement in cholesterol management efforts with the enablers of pharmacy, providers, electronic medical records (EMR), and patients. This paves the way for more future research on cardiovascular and statin regimens from different perspectives.

The above responses are precisely provided in the Introduction

We thank the reviewer for their valuable suggestions and observations

Comment 2: The objectives are not clear and there is no adequate description of the methods. I was left without understanding what really was designed and developed the work and how it was designed.

Author’s Response

Objective: The main objective of the study is to test the clinical hypothesis if an education program on electronic pop-up reminders and the evidence-based algorithm has the potential to improve awareness of cardiovascular disease and diabetes mellitus guidelines of targeting prescribers.

The above responses are precisely provided in the Introduction part.

We thank the reviewer for their valuable suggestions and observations

Comment 3: The results of a questionnaire are presented in the Methods section and also some figures do not make sense and seem to be print screens. 

Author’s Response

            Results of the questionnaire given in the Data Analysis part and the figures representing the clarification over the research problem and the statistics of ASCVD and DM.

We thank the reviewer for their valuable suggestions and observations

Reviewer 3 Report

I highly appreciate the text submitted for review. The subject of the paper is important, however following issues should be addressed. Therefore, I recommend major revision.

1) The paper needs to be proofread for the structure and one more round regarding spelling rules and punctuation.
2a) The research question was not clear. It should be emphasized in the introduction section by clear statements; and
2b) Objective should be clearly explained in the introduction section.
3) Review the methodology section: this reviewer suggests including the time framework and ethical and statistical considerations.

4)  Literature gap and how this paper could fill the gap is not clear because of the lack of a proper literature review. It should be addressed (and reference 15 deleted).

Author Response

Comment 1: The paper needs to be proofread for the structure and one more round regarding spelling rules and punctuation.

Author’s Response

Yes, paper went out for one more round of proof reading.

We thank the reviewer for their valuable suggestions and observations

Comment 2: The research question was not clear. It should be emphasized in the introduction section by clear statements; and Objective should be clearly explained in the introduction section.

Author’s Response

Approximately 945,000 deaths are attributed to cardiovascular diseases annually in the United States, which accounts for about 41% of all mortalities in the country. Cardiovascular diseases also contribute to high morbidity in the country. Indeed, they account for about 6 million hospital admissions annually. Diabetes mellitus refers to dysglycaemia that is accompanied by nephropathy, retinopathy, and cardiovascular diseases, among others. Some of the cardiovascular diseases that are associated with diabetes mellitus include congestive heart failure, stroke, cardiomyopathy, and coronary heart disease, among others. cardiovascular disease and diabetes mellitus comorbidities is associated with early death. In 1999, a collective effort of the following organizations resulted in the recognition of the need for collective cooperation in the prevention of cardiovascular disease in diabetes patients: the American Diabetes Association (ADA), the National Institute of Diabetes and Digestive and Kidney Diseases, and National Heart, Lung, and Blood Institute (NHLBI), Juvenile Diabetes Foundation International, and the American Heart Association (AHA). The pathophysiology of cardiovascular disease involves the thickening of arterial walls, which takes place gradually.

Clinical detection of atherosclerosis is only possible at advanced level, which has made the use of clinical assessment impractical. However, advancement in imaging technology has produced sophisticated and effective tools for detecting atherosclerosis at an early stage. These advancements have also made it possible to classify people on the basis of their exposure to risk of atherosclerosis symptoms. these advancements also provide the means for assessing the treatment results and improving the current understanding of atherosclerosis biology. The problem statement of the present study is as follows. The American College of Cardiology/American Heart Association (ACC/AHA) released new cholesterol management guidelines in 2019. Since then, the guidelines have been updated severally. Indeed, the guidelines are updated frequently to incorporate new evidence. However, in many cases, healthcare providers do not keep up with the frequency at which changes are made. Consequently, they often fail to implement them in clinical practice.  Based on the latest AHA data, appropriate statin regimen was prescribed to only about 46.4% and 30% of diabetes patients and patients with atherosclerotic cardiovascular disease (ASCVD). Charity care clinics can adopt various interventions for safe statin use. Such interventions include the following: educational programs, frequent reminders, support tools, and outcome evaluation.

There is also a need for healthcare providers to educate their patients on the right dose and side effects of statins. They should also regularly check the list of medications that the patients are taking to ensure safety and prevent drug-drug interaction. There is also a need for nursing staff to verify medication compliance and conducting patient counselling on various issues such as the recommended way to administer drugs and report any possible identified concern prescribers. Efforts should also be made to ensure equal accessibility of same information among members of interprofessional teams for informed decision-making and improved therapeutic outcomes. Consequently, we present a hypothesis that an approach that involves systematic identification of interventions that are targeted and barriers at the following levels present high prospects of improving cholesterol management efforts: electronic medical records (EMR), pharmacy levels, and providers.

The main objective of the study is to test the clinical hypothesis if an education program on electronic pop-up reminders and the evidence-based algorithm has the potential to improve awareness of cardiovascular disease and diabetes mellitus guidelines of targeting prescribers. Current study has adopted the Qualitative observation method. Accordingly, the study approached the charity care primary clinic serving a large population situated in the northeastern part of the United States constitutes the setting of the project. The facility has 51 internal medicine residents. The facility has EHR which is used by the clinical staff. Besides, providers use electronic medication prescribing (E-Scribe).

Although a significant number of researches, findings and advancements are located in the treatment strategies for cardiovascular diseases. Its insights into the risk factors of cardiovascular diseases have been evolving over time. Correspondingly, Cardiovascular diseases are still among the leading causes of death in the developed world. According to the reports of the American College of Cardiology/ American Heart Association (ACC/AHA) released new cholesterol management guidelines in 2019, statin regimen was prescribed to only about 46.4% and 30% of diabetes patients and patients with atherosclerotic cardiovascular disease (ASCVD). While Atherosclerotic cardiovascular disease accounts for most deaths and disabilities in North America. This study argues that a systematic approach to identifying targeted interventions to subscribe to the statin regimen to the ASCVD is sparse in the previous studies. This study seeks to address the research gap. Besides, study argues that the Statin regimen could result in improvement in cholesterol management efforts with the enablers of pharmacy, providers, electronic medical records (EMR), and patients. This paves the way for more future research on cardiovascular and statin regimens from different perspectives.

The above responses are precisely provided in the Introduction

We thank the reviewer for their valuable suggestions and observations

Round 2

Reviewer 3 Report

I appreciate the revised manuscript. Some relevant changes have been made, therefore, I recommend to accept in the present form.

Author Response

Hello, thank you for the comments to improve and to accept the final version. I have made some final edits to the abstract and another round of proof-reading. Thank you.
